# Biological and Chemical Characterization of *Musa paradisiaca* Leachate

**DOI:** 10.3390/biology12101326

**Published:** 2023-10-11

**Authors:** Isabelle Boulogne, Philippe Petit, Lucienne Desfontaines, Gaëlle Durambur, Catherine Deborde, Cathleen Mirande-Ney, Quentin Arnaudin, Carole Plasson, Julie Grivotte, Christophe Chamot, Sophie Bernard, Gladys Loranger-Merciris

**Affiliations:** 1Université de Rouen Normandie, Normandie Univ, GlycoMEV UR 4358, SFR Normandie Végétal FED 4277, Innovation Chimie Carnot, IRIB, GDR CNRS Chemobiologie, RMT BESTIM, F-76000 Rouen, France; isabelle.boulogne@univ-rouen.fr (I.B.); gaelle.durambur@univ-rouen.fr (G.D.); cathleen.mirandeney@gmail.com (C.M.-N.); quentin.arnaudin@inrae.fr (Q.A.); carole.plasson@univ-rouen.fr (C.P.); julie.grivotte@laposte.net (J.G.); sophie.bernard@univ-rouen.fr (S.B.); 2Université des Antilles, UMR ISYEB-MNHN-CNRS-Sorbonne Université-EPHE, UFR Sciences Exactes et Naturelles, Campus de Fouillole, F-97157 Pointe-à-Pitre, Guadeloupe, France; philippe.petit@univ-antilles.fr; 3ASTRO Agrosystèmes Tropicaux, INRAE, F-97170 Petit-Bourg, Guadeloupe, France; lucienne.desfontaines@inrae.fr; 4INRAE, PROBE Research Infrastructure, BIBS Facility, F-44300 Nantes, France; catherine.deborde@inrae.fr; 5INRAE, UR1268 BIA Biopolymères Interactions Assemblages F-44300 Nantes, France; 6Université de Rouen Normandie, Normandie Univ, INSERM, CNRS, HeRacLeS US 51 UAR 2026, PRIMACEN, F-76000 Rouen, France; christophe.chamot@inserm.fr

**Keywords:** fungistatic activity, plant elicitor, fulvic acids, organic and inorganic salts, indole alkaloids, PNPP, Basic Substance, SNUB, biostimulant

## Abstract

**Simple Summary:**

There is a growing demand for molecules of natural origin for biocontrol and biostimulation, given the current trend away from synthetic chemicals. Leachates extracted from plantain stems, obtained after the biodegradation of plant material, were characterized to test their potential role as fungicides, plant defense elicitors, and/or plant biostimulants. The plant extracts induced a slight inhibition of fungal growth of an aggressive strain of *Colletotrichum gloeosporioides*, responsible for anthracnose. Organic compounds such as cinnamic, ellagic, quinic, and fulvic acids and indole alkaloids such as ellipticine, as well as minerals such as potassium, calcium, and phosphorus, may be responsible for the inhibition of fungal growth. Jasmonic, benzoic, and salicylic acids have also been found. These are known to play a role in plant defense and as biostimulants in tomatoes. Indeed, foliar application of banana leachate induced overexpression of the *LOXD*, *PPOD*, and *Worky70-80* genes, which are involved in phenylpropanoid metabolism, jasmonic acid biosynthesis, and salicylic acid metabolism, respectively. Leachate also activated root growth in tomato seedlings. However, the main effect of leachate was observed in mature plants, where it reduced leaf area and fresh weight, remodeled stem cell wall glycopolymers, and increased proline dehydrogenase gene expression.

**Abstract:**

There is a growing demand for molecules of natural origin for biocontrol and biostimulation, given the current trend away from synthetic chemical products. Leachates extracted from plantain stems were obtained after biodegradation of the plant material. To characterize the leachate, quantitative determinations of nitrogen, carbon, phosphorus, and cations (K^+^, Ca^2+^, Mg^2+^, Na^+^), Q2/4, Q2/6, and Q4/6 absorbance ratios, and metabolomic analysis were carried out. The potential role of plantain leachates as fungicide, elicitor of plant defense, and/or plant biostimulant was evaluated by agar well diffusion method, phenotypic, molecular, and imaging approaches. The plant extracts induced a slight inhibition of fungal growth of an aggressive strain of *Colletotrichum gloeosporioides*, which causes anthracnose. Organic compounds such as cinnamic, ellagic, quinic, and fulvic acids and indole alkaloid such as ellipticine, along with some minerals such as potassium, calcium, and phosphorus, may be responsible for the inhibition of fungal growth. In addition, jasmonic, benzoic, and salicylic acids, which are known to play a role in plant defense and as biostimulants in tomato, were detected in leachate extract. Indeed, foliar application of banana leachate induced overexpression of *LOXD*, *PPOD*, and *Worky70-80* genes, which are involved in phenylpropanoid metabolism, jasmonic acid biosynthesis, and salicylic acid metabolism, respectively. Leachate also activated root growth in tomato seedlings. However, the main impact of the leachate was observed on mature plants, where it caused a reduction in leaf area and fresh weight, the remodeling of stem cell wall glycopolymers, and an increase in the expression of proline dehydrogenase.

## 1. Introduction

The massive use of phytopharmaceutical products in conventional agricultural systems over several decades has led, among other things, to proven contamination of soil and groundwater, with consequences for human health and consumer doubts about the quality of agricultural food production [1,2]. For example, the strong suspicion of carcinogenicity of the organochlorines used in the West Indian banana industry (chlordecone crisis) was a wake-up call for French West Indians, who are now suspicious of synthetic agrochemicals and, like other populations, are looking for guarantees of the safety of the agricultural food they eat [3]. There is therefore a new demand, or at least a new focus, on agricultural production methods. Thus, it is essential to promote sustainable agricultural systems, i.e., systems that can simultaneously provide food and a variety of ecosystem services [4,5]. 

In recent years, there has been a growing interest in natural preparations of low concern (PNPPs). In France, these preparations are divided into two major categories: Basic Substances (SB) with biocontrol activities and Natural Substances with Biostimulant Use (SNUB) with biostimulant activities [6]. Basic Substances are substances of animal, plant, or mineral origin used for food and feed, useful for plant protection and having no immediate or delayed harmful effects on human or animal health or any unacceptable effect on the environment. They may be used as fungicides, bactericides, insecticides, acaricides, molluscicides, herbicides, nematicides, chemical mediators, or plant defense elicitors. SNUB are non-formulated extracts that may be of animal, mineral, or plant origin. When they are of plant origin, they are plants or algae of the French Pharmacopea or edible parts of plants or algae used for food and feed, obtained by a process accessible to any end user. The effects of these SNUBs applied to plants or their rhizosphere are to improve nutritional efficiency, product quality, and/or tolerance to abiotic stresses [6,7,8,9].

Plant leachates or slurries are PNPPs based on plants and the bioactive compounds released during degradation with living microbes. They have been described by several authors as potential alternatives to replace synthetic phytopharmaceutics [10,11,12]. Interestingly, although *Musa paradisiaca* rachis leachate is a traditional and well-known PNPP treatment in South America and Cuba, very few studies have reported its biocontrol or biostimulant activities [13]. In fact, only some papers have reported a nematicidal activity on *Pratylenchus coffeae* and *Radopholus similis* [14,15], fungicidal activities [16,17,18], or yield increase [19]. 

Thus, the present study focused on the leachate of *Musa paradisiaca* rachis, its chemical characterization, and its use as a potential Basic Substance (antifungal and plant defense elicitor) or SNUB (biostimulant).

As models for this study, we chose *Colletotrichum gloeosporioides* (Penz) Sac., the causal agent of anthracnose, which causes considerable damage to several tropical crops [20,21], and tomato (*Solanum lycopersicum* L.), one of the best studied cultivated dicotyledonous plants, with has important economic and nutritional values [22].

## 2. Materials and Methods

### 2.1. Leachate

Leachate was prepared according to the traditional method with small pieces of *Musa paradisiaca* (French Plantain cultivar) rachis left to decompose in a closed vessel at laboratory temperature, pression, and humidity. The preparation was used after 2 months. The native plantain rachis leachate was collected without any solid material and stored at 4 °C prior to analysis. Freeze-dried leachate was prepared in Guadeloupe from 500 mL of liquid leachate to obtain 60 g of powder (12% dry matter). The freeze-drying was used to find the biological active concentration and standardize the procedure for the assays.

### 2.2. Chemical Analysis

Chemical analyses were performed directly on the liquid phase leachate and after freeze-drying [23]. Total C and total N in the liquid phase and after freeze-drying were determined using an element analyzer (NC 2100 Soil; CE Instruments, Milan, Italy). In 0.1 g of freeze-drying leachate, total P, Ca, Mg, K, and Na were determined after mineralization at 520 °C within one hour and digestion with 37% HCl then solubilized in 25 mL of 10% (*v*/*v*) HCl. Total P was measured colorimetrically according to the method proposed by Novozamsky et al. [24], using a spectrophotometer (Cary 100; Varian Inc., Grenoble, France). Total Ca, Mg, K, and Na were measured by atomic absorption spectrophotometry (AAFS 240; Varian Inc., Grenoble, France). We also analyzed soluble P, Ca, Mg, and Na directly in liquid leachate and in lyophilized leachate after dissolution of 0.1 g of sample in 25 mL of demineralized water, using the same method described above. Quantitative determination of nitrate-N and ammonium-N was performed in liquid leachate and in 0.1 g of freeze-dried leachate dissolved in 25 mL of demineralized water. The measurement was carried out using a continuous-flow colorimeter (AutoAnalyzer 3; Bran+Luebbe, Le Mans, France). The spectroscopic characterization of the humified organic material was carried out with a spectrophotometer (Cary 100; Varian Inc.) at 280, 472, and 664 nm after 24 h extraction of 0.1 g of freeze-dried leachate in 25 mL of 0.5 M NaOH and on liquid leachate after dilution by 10. Absorbance ratios Q2/4 (e.g., A280/A472), Q2/6 (e.g., A280/A664), and Q4/6 (e.g., A472/A664) were then calculated. All chemical determinations were performed in duplicate.

### 2.3. Metabolomic Analysis

Metabolic profiling was achieved by performing liquid-state NMR on native plantain rachis leachate, solid-state NMR on freeze-dried leachate, and by UHPLC-LTQ-Orbitrap MS (liquid chromatography-mass spectrometry, LCMS) on native leachate. One millimeter of native plantain rachis leachate was centrifuged at 10,000× *g* for 5 min, and 600 µL of supernatant was transferred to a NMR tube, with 66 µL of D_2_O and with sodium trimethylsilyl [2,2,3,3-*d*_4_] propionate (TSP, 0.01% final concentration in NMR tube for chemical shift calibration). Liquid state NMR spectra were recorded at 500.162 MHz on a Bruker Avance III spectrometer (Bruker, Wissembourg, France) and using an inverse ATMA broadband 5 mm z-gradient probe at 300 K. A single-pulse sequence with presaturation (zgpr) was used. A total of 512 scans of 64k data points each were acquired with a 90° pulse angle, a 6000 Hz spectral width, a 5.45 s acquisition time, and a 10 s recycle delay. Apodisation (LB 0.2 Hz), zero-filling (X2), and Fourier transformation of Free Induction Decay, phasing, chemical shift calibration, and global and local baseline correction of spectra were carried out using TopSpin 3.6.2 (Bruker, Wissembourg, France). In addition, 1D ^13^C experiments and 2D experiments, ^1^H-^1^H homonuclear correlated spectroscopy (COSY, TOCSY), ^1^H-^13^C heteronuclear single-quantum correlation (HSQC NUS), and ^1^H-^13^C heteronuclear multiple bond correlation (HMBC NUS) experiments were performed to assist in the resonance annotation.

Solid-state ^1^H/^13^C magic angle spinning (MAS) NMR experiments were performed on a Bruker Advance NEO 400 MHz spectrometer by using a double resonance H/X CP/MAS 4 mm probe. The sample was spun at room temperature (298 K) and at a rate of 12 kHz. The cross-polarization pulse sequence parameters were as follows: 2.5 μs proton 90° pulse, 1.5 ms contact time at 54 kHz, and 8 s recycle time. Results were obtained after the accumulation of 2048 scans. The ^13^C NMR chemical shift was calibrated using the carbonyl signal of glycine at 176.03 ppm, as an external standard. Apodisation (LB 50 Hz), zero-filling (X2), and Fourier transformation of Free Induction Decay, phasing, chemical shift calibration, and global baseline correction of spectra were carried out with TopSpin 4.1.4.

Native plantain rachis leachate was subjected to untargeted metabolic profiling by UHPLC-LTQ-Orbitrap MS (liquid chromatography-mass spectrometry (LCMS)) using an Ultimate 3000 ultra-high-pressure liquid chromatography (UHPLC) system coupled to an LTQ-Orbitrap Elite mass spectrometer interfaced with an electrospray ionization (ESI) source (ThermoScientific, Bremen, Germany) operating in both negative and positive ion modes as described in the literature [25,26].

### 2.4. M. paradisiaca Leachate as a Fungicide

Isolate Cg60 was selected from the INRAE culture collections of *C. gloeosporioides* from the French West Indies on the basis of its degree of aggressiveness and rapid growth. The strains were grown on potato dextrose agar medium in a controlled temperature chamber at 23–25 °C. Spores of seven-day old cultures were suspended in distilled water and adjusted to concentrations of 2·10^6^ to 10^7^ conidia/mL using a hemocytometer [27].

An agar well diffusion method was used to screen for fungistatic activity [28,29]. *C. gloeosporioides* inoculum (2 mL at 2·10^6^ to 10^7^/mL conidia/100 mL of medium) were placed in LB agar medium 10 g·L^−1^ at 37 °C and were plated on Petri dishes (80 mm diameter). Using a sterile borer, 6 wells of 2 mm diameter were punched on agar medium, which contained conidia. Each well was filled with 150 µL of the solutions to be tested. Petri dishes were incubated at 26 °C. Fungistop FL (a commercial preparation containing 500 g·L^−1^ of chlorothalonil used against anthracnose; PHYTEUROP, Levallois-Perret, France) was used as standard fungicidal agent, and sterile distilled water as negative control. Results were recorded after 24 h and one week. Each condition was replicated 10 times. The diameter of inhibition fungal growth was measured. 

Four sterilization procedures were tested to prevent microbial contamination. Leachate and controls (e.g., Fungistop FL and distilled water) were filtered on 0.2 μm Millipore filters, supplemented with streptomycin at 200 µg·mL^−1^, autoclaved, or UV sterilized. The fungistatic test with leachate after freeze-drying with streptomycin at 200 µg·mL^−1^ was performed with 4 concentrations of freeze-dried leachate (0.5, 1, 2, and 5 mg·mL^−1^). 

### 2.5. M. paradisiaca Leachate as a Plant Defense Elicitor

*Solanum lycopersicum* L. seeds (cv. St Pierre) were sterilized with ethanol and sodium hypochlorite and cultivated on agar ½ MS medium at 24 °C (80–90% relative humidity (RH), 16- to 8-h day/night cycle), in a growth chamber for 13 days. Three days after germination, seedlings were treated with water or 5 mg·mL^−1^ (*w*/*v*) freeze-dried leachate. Each condition was replicated in 4 independent biological replicates with 20 seedlings in each replicate.

Gene expression analysis was performed on total RNAs extracted from 13-day-old seedlings grown under control (water) or leachate-spray conditions (5 mg·mL^−1^ freeze-dried leachate) using the Macherey Nagel NucleoSpin RNA Plant kit. The reverse transcription was performed from 1 μg of total RNAs using the High-Capacity cDNA Reverse Transcription Kit. The reaction mix was prepared in a 96-well reaction plate using the Fast SYBR Green Master Mix (Applied Biosystems) in a final volume of 13 μL with 200 nM of each primer and 3 μL of cDNA template. The selected genes were amplified using real-time PCR: *LOXD* (forward primer 5′ CGGAGAGTCGTGTCGAGA 3′, reverse primer 5′ TTAAGCCTGGAGGTTGAGAATG 3′), *PAL-5* (forward primer 5′CGGTGTGACTACTGGATTTGG 3′, reverse primer 5′ CTGCCCTTGTTGCTGAATGT 3′), *PPO-D* (forward primer 5′ GGCTTAGGAGGTCTTTATGGTG 3′, reverse primer 5′ ATCAGGAGGTGGTGTAGGAG 3′), *Pti-5* (forward primer 5′ ATTCGCGATTCGGCTAGACATGGT 3′, reverse primer 5′ AGTAGTGCCTTAGCACCTCGCATT 3′), *Worky28* (forward primer 5′ ACAGATGCAGCTACCTCATCCTCA 3′, reverse primer 5′ GTGCTCAAAGCCTCATGGTTCTTG 3′), *Worky70-80* (forward primer 5′ GGGCCAGATCGAGGAAGTTG 3′, reverse primer 5′ GCCCATATTTTCTCCATGCACA 3′). Two others genes were used as reference: *Ef1* (forward primer 5′ GGAACTTGAGAAGGAGCCTAAG 3′, reverse primer 5′ CAACACCGACAGCAACAGTCT 3′) and *PHD* (forward primer 5′ ATTCGTGGCTGCTCTCTGTC 3′, reverse primer 5′ CCCTGTCACGGCTTCAAAGA 3′). Primers were designed using (1) a BLAST of the homology transcript sequences of two other plant species (*Arabidopsis thaliana* and *Populus trichocarpa*) on the tomato genome; (2) the use of NCBI’s Primer-BLAST tool; and then (3) produced by Kaneka Eurogentec SA.

Quantitative real-time PCR was performed using the CFX96 real-time system (Bio-Rad). The following parameters were used: 20 s at 95 °C then 40 cycles of 5 s at 95 °C, 20 s at 60 °C followed by the melt-curve analysis: 15 s at 95 °C, 6 min at 60 °C, and 15 s at 95 °C. Data were analyzed using the CFX Maestro software (Bio-Rad, Hercules, CA, USA). Relative expression was normalized using the 2−∆∆Cq method to Ef1 and PHD as reference genes using GENorm analysis (provided in CFX Maestro software 2.3 version).

### 2.6. M. paradisiaca Leachate as a Biostimulant

For the in vitro assay, *S. lycopersicum* L. seeds (cv. St Pierre) were sterilized with ethanol and sodium hypochlorite and grown on agar ½ MS medium at 24 °C (80–90% relative humidity (RH), 16- to 8-h day/night cycle), in a growth chamber. Seedlings were treated with water or freeze-dried leachate (5 mg·mL^−1^ (*w*/*v*)) three days after germination and transferred on agar-gelled media, infused 16 h with 50% PEG_6000_ seven days later to simulate osmotic stress. Harvesting was performed after a total of 13 days. Each condition was replicated in 3 independent biological replicates with 90 seedlings in each replicate. Indicators of biostimulatory activity in the in vitro assay are (1) measurements of aerial and root parts, (2) fresh and dry biomass, (3) water content, (4) measurement of destructive chlorophyll content, (5) gene expression analysis, and (6) cell wall imaging.

For the phytotron assay, seeds were sterilized with ethanol and sodium hypochlorite and cultivated on Jiffy Growblock at 24 °C (60% relative humidity (RH), 16- to 8-h day/night cycle), in a Strader^®^ Phytotron for 45 days with irradiance provided by LEDs with growth/flowering spectrum AP67. The plants were foliar sprayed twice with water or leachate (5 mg·mL^−1^ (*w*/*v*) of freeze-dried leachate) 17 and 24 days after sowing. Drought stress was applied 31 days after sowing for a period of 14 days at FC25% (e.g., with a water application restriction of 75% of the substrate field capacity). Harvesting took place 45 days after sowing. Each condition was replicated in 4 independent biological replicates with 15 plants in each replicate. Indicators to demonstrate biostimulatory activities in the phytotron assay are (1) aerial parts, leaf surface and stem diameter measurements, (2) fresh and dry biomass, (3) aerial parts water content, (4) non-destructive and destructive chlorophyll content measurements, (5) gene expression analysis, and (6) cell wall imaging.

(1) For the in vitro assay, aerial and root parts were measured after photography using the ImageJ software (1.54f version, NIH). For the phytotron assay, aerial parts were measured with a measuring tape, stem diameter with an electronic caliper, and leaf surface after scanning using the ImageJ software. 

(2) In both assays, fresh and dry biomasses were obtained by weighing on a balance before and after drying at 60 °C for 7 days. 

(3) Water content in seedlings or in aerial parts was calculated by subtracting dry mass from fresh mass. 

(4) For both assays, destructive measurements of chlorophyll content were carried out using the Arnon method. First, 150 mg of fresh aerial parts of seedling or leaves of mature plants were homogenized in 80% acetone, centrifuged, and the absorbance of the supernatant was read at 645 nm and 663 nm against the blank. The amount of chlorophyll present in the sample was calculated using Arnon’s formula: Chl a (mg·g^−1^) = [(12.7 × A663) − (2.6 × A645)] × mL acetone/mg leaf tissue; Chl b (mg·g^−1^) = [(22.9 × A645) − (4.68 × A663)] × mL acetone/mg leaf tissue; Total Chl = Chl a + Chl b. For the phytotron assay, non-destructive measurements were made on the youngest fully expanded leaves using a chlorophyll meter (SPAD-502, Minolta, Japan). 

(5) Gene expression analysis was performed on total RNAs extracted from 13-day-old seedlings for the in vitro assay and from leaves of 45-day-old plants for the phytotron assay using the same methodology as described in Section 2.5. The genes selected were: *P5CS* (forward primer 5′ TGATGGAAGATTAGCACTTGGAA 3′, reverse primer 5′ CAACACCTACAGCACCAGAA 3′), *ProDH* (forward primer 5′ TCGTGCGCGGAGTTTATGAT 3′, reverse primer 5′ GGTTGCAGCAAGTTTTCCTGA 3′), and AREB1 (forward primer 5′ GGACTTGGGAAGAGCAATGGA 3′, reverse primer 5′ AGCTCCATAGTATAGGCCTGCT 3′). The reference genes were also Ef1 and PHD.

(6) Cell wall imaging was performed using resin embedding and immunofluorescence labeling. For this purpose, 0.5 cm long sections were collected from the stem just below the hypocotyl of the 13-day-old seedling for the in vitro assay, and from the stem cross-section below the youngest fully expanded leaves for the phytotron assay. Sections from three seedlings or plants for each condition were processed at 4 °C using an electron microscopy tissue processor (EM-TP, Leica microsystems) as follows. Samples were fixed for 1 h 30 in 1% paraformaldehyde and 1% glutaraldehyde mixture (v/v) in 0.1 M sodium cacodylate buffer solution (pH 7.2) and washed (4 × 5 min) in ultrapure water. The samples were then dehydrated in an ethanol series (30%, 50%, 70%, and 2 × 100%) 2 h each and embedded in LRW resin through a LRW-ethanol series (25%, 50%, 75%, and 100%), 24 h each. Finally, the samples were embedded (6 × 24 h) in LRW resin complemented with the UV catalyst benzoin methyl ether (0.5% w/v) and polymerized with UV light for 48 h at 4 °C. Sections from resin blocks (2 µm; EM UC6 Leica microsystems) were collected on 10 well slides previously coated with poly-L-lysine (0.01% v/v). Resin sections were blocked for 30 min in PBS-Tween 20 0.1% (w/v)) supplemented with 3% of BSA (bovine serum albumin) and NGS 1/20 (normal goat serum, v/v). Sections were washed in PBS-T + 1% BSA (5 × 5 min) and incubated overnight at 4 °C in a wet chamber with the primary antibody (see list in Appendix A, Plant probes). After washing in PBS-T + 1% BSA (5 × 5 min), sections were incubated for 2 h at 25 °C in a humid chamber with a rat or mouse secondary antibody coupled to Alexa 488 (d: 1/200, In Vitrogen). Finally, sections were washed in PBS-T + 1% BSA (5 × 5 min) and mQ water (2 × 5 min). Fluorescence on sections was observed using a macroscope Axiozoom Zeiss, with a filter set 38 HE (BP 470/40 BP 525/50), an exposure time of 400 ms, a range of 50 µm, slides: 11, intervals: 5 µm. Negative controls were performed by omitting the primary antibody. 

### 2.7. Image Analysis and Statistical Tests

For in vitro assay images, fluorescence measurements were performed on the region of interest (ROI) containing all the tissues of the stem and used to obtain a mean of fluorescence intensity value of the ROI. For phytotron assay images, fluorescence measurements were performed on four zones containing different tissues of the stem (zone 1: epidermis + peridermis + collenchyma; zone 2: cortical parenchyma; zone 3: sclerenchyma + xylem + phloem; zone 4: medullar parenchyma) (Appendix A). To automate the measurements, an ImageJ macro was developed (Appendix A). It manages the opening of an image, the projection of average intensity along the Z axis, and the retrieval and saving of intensities along a line in a results table. Z-average projection is used to smooth out cutting effects, as each point on the line is itself an average of a user-defined orthogonal segment.

Data were analyzed by non-parametric analyses using the Kruskal–Wallis test with multiple comparisons of the Dunn method and the Mann–Whitney test using XLSTAT^®^ software 2021.3.1.

## 3. Results and Discussion

### 3.1. Chemical Composition of Leachate

Chemical analyses of the liquid leachate showed that the three major elements were potassium (33.61% MS), phosphorus (0.51% MS), and calcium (0.39% MS) (Table 1). These elements were found in solution and were therefore not associated with any organic matter. On the other hand, magnesium and sodium were present in small amounts. 

For freeze-dried leachate, Table 1 showed that the most important element remained potassium with 29.57%. Nitrogen and carbon are the other two. In contrast to the liquid leachate, phosphorus decreased in freeze-dried leachate and became 0.06%.

Absorbance ratios for the liquid leachate were 6.60, 23, and 3.50 for Q2/Q4, Q2/Q6, and Q4/Q6, respectively. Freeze-dried leachate had a lower absorbance (6.60, 43.42, and 6.40, respectively) (Table 2). 

The Q2/4 reflects the proportion between lignins and quinones (material at the very beginning stage of transformation) and the organic matter at the beginning of humification. Q2/6 is the ratio between non-humified and highly humified material. Finally, the Q4/6 ratio indicates the relationship between material at the beginning of humification and highly humified material with a high degree of aromatic condensation [30,31]. Furthermore, a low Q4/6 ratio indicates a high degree of humification [30] and fulvic acids are mostly involved when it is higher than 5 [32]. Thus, our leachate is composed of non-humified and highly humified material due to its high Q2/6 ratio. In addition, its low level of Q4/6 ratio less than 5 indicates that fulvic acids are mostly involved. This is in line with the literature on *Musa paradisiaca* leachate, which indicates its significant fulvic acid content [33]. 

In terms of chemical analysis, we showed that potassium was the most important element in the *Musa parasidiaca* leachate. Velez and Zapata [33] also reported a high concentration of this alkali metal in their fulvic acids extracted from *M. parasidiaca* rachis. 

In addition to potassium, phosphorus and calcium were also notable soluble elements in our liquid leachate. This result was in line with another study, which dealt with the effect of leachates from rachis of several *Musa* species (Cavendish and Plantain cultivars) on *Mycosphaerella fijiensis* (causal agent of Black sigatoka). They showed that Cavendish and Plantain leachates contained phosphorus (0.24 to 0.61%), potassium (218 to 8169 mg·L^−1^), and calcium (13.25 to 652.5 mg·L^−1^) [34]. In our leachate, we obtained results in the same concentration range as those previously obtained with 0.51%, 3664 mg·L^−1^, and 42.48 mg·L^−1^, respectively, for the same elements. 

### 3.2. Metabolomic Analysis

Analysis of native plantain rachis leachate by ^1^H liquid-state NMR revealed the presence of an intense and large singlet at 1.92 ppm (HSQC cross peak 24.8 ppm) and a series of small resonances, including a doublet at 0.91 (HSQC cross peak 24.5 ppm), a triplet at 1.06 ppm (HSQC cross peak 13 ppm), a singlet at 1.07, a quadruplet at 2.18 ppm (HSQC cross peak 33.3 ppm), a quintuplet at 2.39 ppm, singlets at 3.36, 3.43, and 3.54 ppm, and characteristic resonances of aromatic protons, five multiplets at 7.88, 7.56, 7.49, 7.31, and 7.39 ppm, and two doublets at 6.82 and 7.14 ppm (Appendix A).

The TOCSY cross peaks at 6.87/7.06 ppm confirm a C-C bond for the latter two doublets. Comparison of the leachate ^1^H NMR spectrum with the benzoic acid one obtained in deuterated phosphate buffer solution (200 mM, apparent pH 6.0) at 500 MHz showed similar NMR patterns and intensity ratios between the benzoic acid signals and the three multiplets observed at 7.88, 7.56, and 7.49 ppm (Appendix A). 

The ^13^C liquid-state NMR spectrum showed three main resonances, a thin one at 26.1, a large one at 165.0 ppm, and a thin one at 184.3 ppm. The HMBC spectrum confirmed the presence of acetyl-group (HMBC cross peak 26.2/185 ppm) (Appendix A). 

The dry matter content of the native plantain rachis leachate was 12%. The analysis of this freeze-dried leachate powder by solid-state ^1^H/^13^C MAS NMR revealed at least two mobile components at 161 and 41.5 ppm and several rigid components, with different types of chemical structures putatively annotated according to Preston et al. [35]. The mobile component at 161 ppm is probably due to polyphenolic compounds (Appendix A). NMR metabolomic analysis also revealed that both native and freeze-dried leachate were rich in polyphenolic and carboxylic acid derivatives, which was in full accordance with untargeted profiling LCMS experiments. Raw LCMS data were processed using in-house optimized parameters yielding 1200 metabolomic signals [25]. We then conducted a tentative annotation of the nine metabolic compounds that may explain the targeted activities. A putative prediction of compounds indicated molecules implicated in plant defense, plant growth metabolism, tolerance to abiotic stress, and antioxidant and antimicrobial activities [36,37,38]. Indeed, cinnamic acid, ellagic acid, quinic acid, ellipticin, chalcones, retrochalcones, and fulvic acids along with jasmonic, benzoic, and salicylic acids were annotated. It is thus reasonable to first explore the potential fungistatic effect of our leachate.

### 3.3. M. paradisiaca Leachate as a Fungistatic Agent

As axenic conditions are required to correctly screen fungistatic activity, the diameter of inhibition of *C. gloeosporioides* growth was measured on the leachate prepared according to four procedures of sterilization. Our results showed that liquid leachate without sterilization inhibited fungal growth compared to water control after 24 h. Moreover, among the sterilization procedures, only the sterilization procedure with streptomycin (antibacterial which inhibits protein synthesis) still maintained an effect on the fungus. After 7 days, all the leachates lost their ability to reduce fungal growth (Table 3).

Following this result, a further fungistatic test was carried out using leachate supplemented with streptomycin at 200 µg·mL^−1^. Four concentrations (0.5, 1, 2, and 5 mg·mL^−1^) of freeze-dried leachate were used. Our results showed that freeze-dried leachate added with streptomycin (at 0.5, 1, 2, and 5 mg·mL^−1^) inhibited fungal growth compared to sterile water control after 24 h (Figure 1). However, the growth inhibition of Fungistop FL was more important than the inhibition of the leachate. After one week, the growth inhibition of Fungistop FL persisted through the time while the fungi recolonized the inhibition zones of the leachates.

Regarding sterilization procedures, Contreras-Blancas et al. [39] showed that leachate produced by composting organic wastes with earthworms had an antifungal activity against *Colletotrichum gloeosporioides* and that sterilization of this leachate strongly reduced the inhibitory effect. We also saw the same effect on our leachate, which was less effective after the addition of streptomycin and which had no more effect after sterilization by UV, microfiltration unit, or autoclaving. Our biopesticide based on microorganisms or their bioactive compounds could be inhibited by any sterilization process. The loss of fungistatic activity of the lixiviate after sterilization by 0.22 µm filtration could be explained by retention of microorganisms, but also by adsorption losses of chemical compounds on membrane filters, both with fungistatic activity. Some compounds, such as salicylic acid and benzoic acid, were reported to have significant adsorption losses on membrane filters [40].

Regarding the fungistatic effect, Dekker and Medlen [41] showed in their patent the same effect of fulvic acids on *Aspergillus niger* and *Candida albicans*. Our leachate with its fulvic acids contents probably have the same fungistatic effect on this *C. gloeosporioides* strain, which explains its temporary inhibition. This lack of persistent activity could be a positive characteristic from the perspective of biodegradability and ecological suitability. Moreover, Glare et al. [11] pointed out that the application of biopesticides based on living microbes and their bioactive compounds rarely has an effect for more than a few days. The observed fungistatic effect may also be related to the presence of cinnamic acid, ellagic acid, quinic acid, ellipticin, chalcones, and retrochalcones, which are known to exhibit antifungal activities [42,43,44,45,46,47].

Lastly, Lemar et al. [48] showed that fresh garlic extract was more effective than freeze-dried garlic extract on morphology and inhibition of fungus growth. Our finding was the same with the freeze-dried leachate, which was less effective than liquid leachate. The microbiota of the leachate is unknown, and it can be assumed that some of them with antifungal activity might be sensitive to freeze-drying and be inactivated. Since the leachate has only fungistatic activity, we decided to investigate the indirect effect on fungal diseases through possible stimulation of plant defense mechanisms. Indeed, Deliopoulos et al. [49] indicated that inorganic salt could be useful in integrated fungal disease management by stimulating dehydration of fungal spores, inhibition of sporulation or mycelium growth, or by stimulating plant defense mechanisms.

### 3.4. M. paradisiaca Leachate as a Plant Defense Elicitor

Following the previous results, we used the freeze-dried leachate at 5 mg·mL^−1^ to investigate gene expression analysis of six key genes in plant defense in tomato: *LOXD* (a lipoxygenase D in jasmonic acid pathway), *PAL5* (Phenylalanine ammonia-lyase in phenylpropanoid pathway), *PPO-D* (a polyphenol oxidase D in phenylpropanoid pathway), *Pti-5* (a pattern-triggered immunity marker), and *Worky28* and *Worky70-80* (respectively Pti marker and transcription factor defense response—salicylic acid related).

Leachate appears to have plant defense elicitor activity by inducing overexpression of *LOXD* (2.80-fold compared to control water), *PPOD* (6.34-fold compared to control water), and *Worky70-80* (1.54-fold compared to control water), which are involved in phenylpropanoid metabolic pathways, jasmonic acid biosynthesis, and salicylic acid metabolism, respectively (Table 4).

It appears that no elicitor effect of *Musa* leachate on tomato has been reported in the literature to date. However, our results are consistent with a previous study where anaerobic fermentation of manure and plant waste (containing humic substances) stimulated salicylic and jasmonic pathways in banana cultivar [50]. Another study showed that fulvic acid regulates phenylpropanoid metabolism in grapefruit [51].

As mentioned above, according to European regulations, biocontrol activities are different from biostimulant activities. However, physiologically, there are similar responses of plants to humic substances in case of biotic or abiotic stress [52]. Furthermore, there are a number of examples showing the potential of humic and fulvic acid based biostimulants to improve abiotic stress tolerance in plants [7]. Therefore, it seemed advisable to test the effect of the leachate as a biostimulant that could potentially improve abiotic stress tolerance.

### 3.5. M. paradisiaca Leachate as a Biostimulant

The potential improvement of abiotic stress tolerance was evaluated in vitro and in phytotron assays on tomato seedlings and mature plants grown under severe drought stress after foliar treatment with freeze-dried leachate at 5 mg·mL^−1^ using phenological, tissular, and molecular tools.

Regarding phenological tools, our results showed that leachate induced a decrease in root length in tomato seedlings, which correlated with an increase in aerial parts/root parts, but an increase in root growth rate (Figure 2A). It is well known that in drought stress, the root length was increased [53]. Thus, the reduction of root length is a positive effect of application of the leachate as a biostimulant indicating a reduction of the stress perceived by the plant.

In mature plants, only decreases in leaf area and fresh weight were observed (Figure 2B).

For the tissular tool, we investigated the immunolocalization of plant glycopolymers epitopes, as there is clearly a close relationship between cell wall glycomolecules and abiotic stress [54]. Indeed, drought stress affects cell wall composition in terms of pectins, hemicelluloses, and hydroxyproline rich glycoproteins (HRGP) [55,56]. Thus, our results showed that under drought stress conditions, application of *Musa* leachate induces remodeling of glycopolymers labeling only in mature tomato stems. Indeed, we observed an increase of pectin (LM5), hemicellulose (LM15), and HRGP (JIM13) in the peripheral zone of the stem (epidermis, peridermis, and collenchyma). Interestingly, for the same JIM13 labeling, we obtained a decrease in cortical parenchyma. However, some other studies on AGP localization in tomato reported that there is a regulation of AGPs at the organ, tissue, and cellular levels [57]. 

In the vascular system zone (sclerenchyma, xylem, and phloem), the leachate seemed to induce an increase in other types of pectins (homogalacturonans) and HRGP (extensins). Finally, we did not observe any cell wall remodeling as a result of leachate application in the medulla of mature tomato stems or in tomato seedlings (Figure 3, Supplemental Appendix A).

Regarding the molecular tool, we investigated the gene expression analysis of three key genes in drought stress metabolism in tomato: *P5CS* (a delta-1-pyrroline-5-carboxylate synthase involved in proline synthesis), *ProDH* (a proline dehydrogenase involved in proline catabolism), and *AREB1* (ABA-responsive element binding protein 1, a transcriptional activator involved in drought stress tolerance).

Our results showed that the leachate increases the expression of proline dehydrogenase gene only in mature plants in drought stress condition (Table 5).

Our results clearly showed that foliar application of *M. paradisiaca* leachate induced a biostimulant effect in stimulating drought stress tolerance in tomato seedlings and mature plants. In young plants, the leachate only induced a reduction in root length and a stimulation in root growth rate compared to control plants under drought stress. Previous studies also observed that when applied to young plants, humic substances appear to have a greater effect on roots than on the aboveground parts of plants [58]. The effect of the leachate was mainly on mature plants, causing a reduction in leaf area and fresh weight, remodeling of plant cell wall glycopolymers, and an increase in the expression of proline dehydrogenase gene.

It has been reported that one of the strategies of drought resistant tomato cultivars could be to have a significantly reduced leaf area in order to reduce the evaporative leaf surface [59]. Therefore, it can be assumed that leachate induced an increase in drought resistance and consequently a reduction in leaf area (correlated with a reduction in aerial fresh weight) in tomato.

Little is known about the effect of humic substances on cell wall remodeling. However, a previous transmission electron microscopy (TEM) study showed that untreated wheat xylem vessels had thinner cell walls compared to those treated with humic substances, which could have a relevant effect on water conductivity and nutrient flux intensity [60]. Furthermore, it is known that there is an increase in pectic polymers (rhamnogalacturonan I and homogalacturonans) [61], xyloglucan [62], or AGPs [55] in drought-tolerant cultivars or species compared to susceptible ones. Extensins are also thought to be involved in drought stress tolerance by cross-linking with the other cell wall components to increase the tensile strength of plant cells [63]. Again, the results suggest that leachate application was able to induce drought tolerance mechanisms.

Finally, it is known that accumulation of proline (by overexpression of *P5CS* and suppression of *ProDH*) improves tolerance to drought stress in various plants [64]. However, since proline is toxic to plants, once the stress is relieved, *ProDH* is activated to degrade the stored proline [65]. Thus, our surprising result for *ProDH* overexpression may indicate that the leachate allowed the tomato plants to recover sufficiently from the water shortage to activate *ProDH* gene.

Regarding the metabolomic profiling obtained, both jasmonic acid and salicylic acid were annotated. In abiotic stress, jasmonic acid is usually involved in physiological and molecular responses. A recent study showed that exogenous application of jasmonic acid has a regulatory effect on plants [66]. Moreover, jasmonic acid does not play an independent regulatory role, but works in a complex signaling network with other phytohormone signaling pathways and has synergistic effects with salicylic acid in the process of resisting environmental stress [66,67,68]. Another study showed that the exogenous application of benzoic acid, which is a precursor of salicylic acid, induced the formation of the latter [69].

## 4. Conclusions

Our study contributes significantly to the knowledge on *Musa* sp. rachis leachate as biocontrol and biostimulant method. Like previous studies, our work showed that *M. paradisiaca* leachate contains humic and fulvic acids and a high concentration of potassium, as well as cinnamic acid, ellagic acid, quinic acid, ellipticin, jasmonic acid, benzoic acid, salicylic acid, chalcones, and retrochalcones. We first observed a slight in vitro fungistatic activity (against an aggressive strain of *C. gloeosporioides*) of the preparation. Subsequently, other investigations performed seemed to indicate that the reduction of disease observed in the field after treatment with this traditional natural preparation of low concern was rather due to the stimulation of plant defense mechanisms and drought stress tolerance.

## Figures and Tables

**Figure 1 biology-12-01326-f001:**
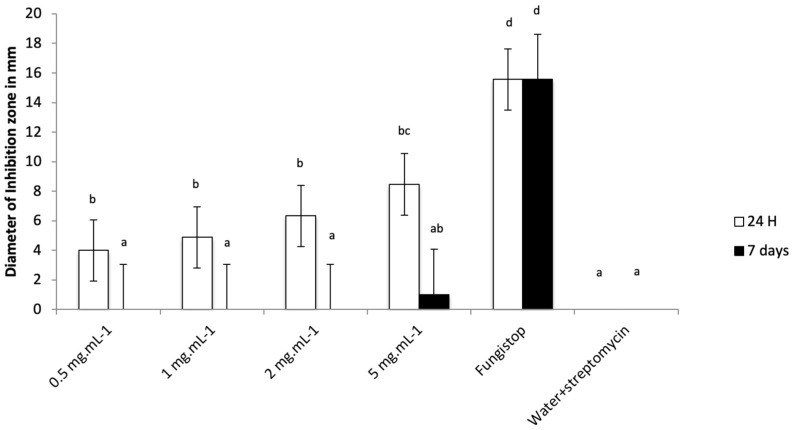
Diameter of inhibition zone of *C. gloeosporioides* in mm for freeze-dried leachate at 0.5, 1, 2, and 5 mg·mL^−1^ and controls after 24 h and 1 week. Bars represent means and error bars are standard deviation. Treatments without common superscript letters (a, b, c, or d) differ significantly based on the Kruskal–Wallis Test with Dunn’s multiple comparison test.

**Figure 2 biology-12-01326-f002:**
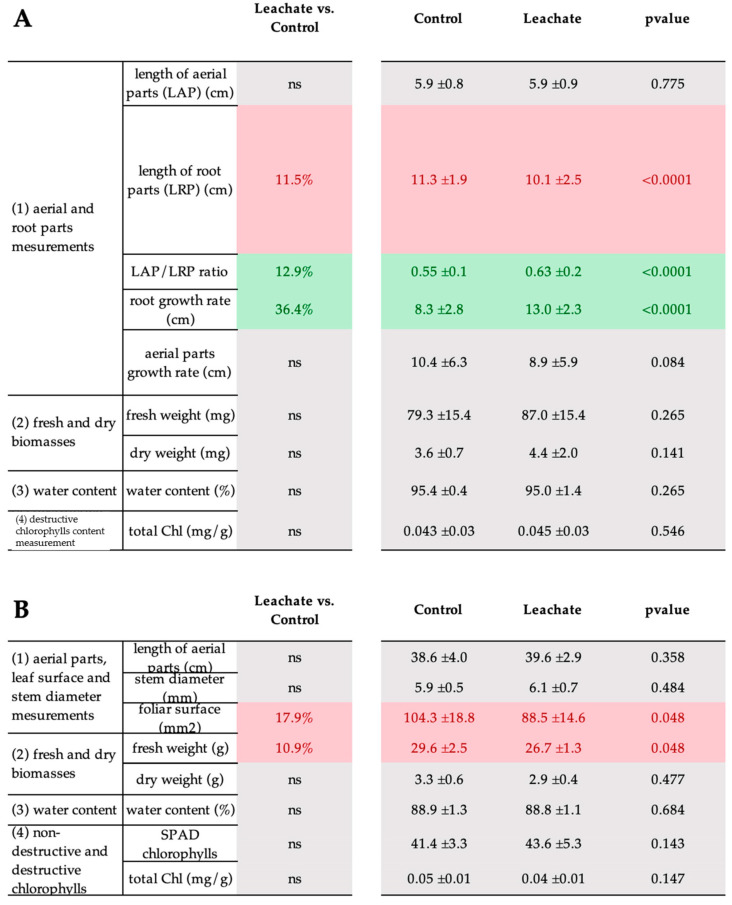
Effect of leachate application on tomato seedlings in in vitro assay (**A**) and mature plants in phytotron assay (**B**) growing under drought stress. The color squares differ significantly based on the Mann–Whitney Test. In green, a significant increase compared to the control; in red, a significant decrease compared to the control; in gray, not significant. Percentages in squares correspond to leachate-induced increase or decrease compared to the control (water). Values on the right correspond to the different measurements and the *p*-value according to the Mann–Whitney Test. For the in vitro assay, each condition was replicated in three independent biological replicates with 90 seedlings in each replicate. For phytotron assay, each condition was replicated in four independent biological replicates with 15 plants in each replicate. ns: not significant.

**Figure 3 biology-12-01326-f003:**
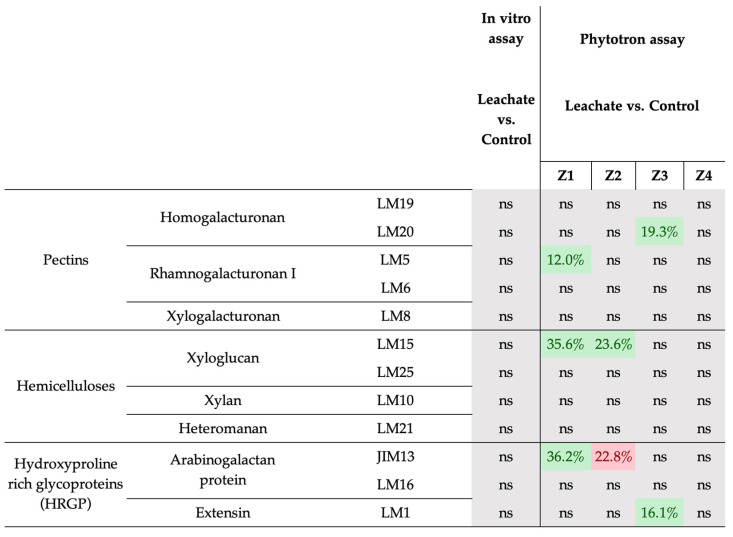
Effect of leachate application under drought stress on cell wall imaging in in vitro and phytotron assays. The color squares differ significantly based on the Mann–Whitney Test. In green, a significant increase compared to the control; in red a significant decrease compared to the control; in gray, not significant. Percentages in squares correspond to leachate-induced increase or decrease in glycopolymers epitopes detection. (Z1) zone 1: epidermis + peridermis + collenchyma; (Z2) zone 2: cortical parenchyma; (Z3) zone 3: sclerenchyma + xylem + phloem; (Z4) zone 4: medullar parenchyma. Each condition was replicated was replicated in three independent biological replicates. ns: not significant.

**Table 1 biology-12-01326-t001:** Quantitative determination of total nitrogen and nitrogen forms, total carbon, phosphorus, potassium, calcium, magnesium, and sodium in g/% dried weight for liquid and freeze-dried leachate.

	Dried Weight	N	C	P	K	Ca	Mg	Na	N-NH4	N-NO3
Liquid leachate	1.09	0.008	0.30	0.51	33.61	0.39	0.009	0.031	0.041	0.00
Freeze-drying leachate	-	0.731	16.04	0.06	29.57	0.07	0.00	0.14	0.12	0.00

**Table 2 biology-12-01326-t002:** Spectroscopic characterization of humified organic materials for liquid and freeze-dried leachate at 280, 472, and 664 nm and absorbance ratios (Q2/4 = A280/A472, Q2/6 = A280/A664, and Q4/6 = A472/A664). OD: optical density.

	OD 664	OD 472	OD 280	Q2/4	Q2/6	Q4/6
Liquid leachate	3.56	12.50	82.50	6.60	23	3.50
Freeze-drying leachate	0.12	0.76	6.06	6.60	43.42	6.40

**Table 3 biology-12-01326-t003:** Effect of the four procedures of sterilization and controls on growth of *C. gloeosporioides* after 24 h and 7 days. + represents the inhibition of the fungus growth and − the lack of inhibition.

	No Sterilization		With Streptomycin (200 µg·mL^−1^)		Sterilization with 0.2 μm Filter		Autoclaved		UV Sterilization	
	24 h	7 days	24 h	7 days	24 h	7 days	24 h	7 days	24 h	7 days
Liquid leachate	+	−	+	−	−	−	−	−	−	−
Fungistop FL	+	+	+	+	+	+	−	−	+	+
Distilled water	−	−	−	−	−	−	−	−	−	−

**Table 4 biology-12-01326-t004:** Gene expression analysis after treatments with freeze-dried leachate at 5 mg·mL^−1^ and water (Control). Genes are expressed by fold change with in bold and asterisk those significantly expressed (≤−1.5 or FC ≥ 1.5; *p*-value ≤ 0.05).

	*LOX-D*	*PAL5*	*PPO-D*	*Pti-5*	*Worky28*	*Worky70-80*
Leachate vs. Control (water)	**2.80 ***	1.37	**6.34 ***	1.45	−1.30	**1.54 ***

**Table 5 biology-12-01326-t005:** Gene expression analysis after treatments with freeze-dried leachate at 5 mg·mL^−1^ and water (Control) for in vitro and phytotron assay. Genes are expressed by fold change with in bold and asterisked those significantly expressed (≤−1.5 or FC ≥ 1.5; *p*-value ≤ 0.05).

	*P5CS*	*ProDH*	*AREB1*
Leachate vs. Control (water) (in vitro)	1.4	1.35	−1.15
Leachate vs. Control (water) (phytotron)	−1.25	**4.45 ***	1.13

## Data Availability

NMR data and metadata have been deposited in the recherche.data.gouv.fr repository (https://doi.org/10.57745/CN44I4 (accessed on 1 September 2023)). Other data are available from the authors upon reasonable request.

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
