# Peer review of "Biological and Chemical Characterization of Musa paradisiaca Leachate"

_biology, 2023, doi:10.3390/biology12101326_

Round 1
Reviewer 1 Report
1. The methodology for leachate preparation appears to be lacking. It does not clearly outline the conditions for digestion. Is the closed vessel the only method used? What was the humidity level during fermentation? Does the leachate contain solids? In my opinion, the use of freezedrying is not recommended as it increases production costs and complicates scaling. Your results indicate that when you lyophilize the sample, the concentrations of nitrogen, potassium, phosphorus, and calcium decrease. If freeze-drying is necessary, a more thorough explanation of the rationale behind this choice is required. 2. It is crucial to standardize the leachate preparation process. Consider conducting a statistical design to determine the variables that have the most significant impact on leachate quality. Since your article is focused on leachate, meticulous attention to the conditions is essential, as these conditions can significantly affect your results. 3. There is no reference to the conditions for in vivo experiments in tomato seedlings. 4. Regarding the gene analysis, did you design the primers, and if so, what criteria guided your primer selection and analysis conditions? Additionally, please provide information about the primer manufacturer. 5. Your leachate does not exhibit fungicidal activity, contrary to your declaration. It is evident from your findings that after 7 days, fungi growth resumes. To substantiate claims of fungicidal activity, additional antagonism tests against different phytopathogenic microorganisms are needed, ensuring that the effect remains consistent over time. 6. A thorough discussion is needed to explain the reasons behind the fluctuations in nitrogen, carbon, potassium, and phosphorus levels. 7. The article should delve into the types of phenolic compounds that may be present in the leachate. Consult existing literature to identify previously reported compounds and their mechanisms. Confirm these predictions through Mass Spectrometry (MS) analysis. 8. Please provide an explanation as to why leachate loses its fungicidal activity when sterilized using a 0.22 µm filter. Additionally, discuss the stability of liquid leachate compared to freezedried leachate. 9. It is worth noting that your leachate negatively affects the root length of tomato seedlings, which may impact the mobilization of nutrients and minerals. Consider conducting viability experiments in a greenhouse setting to further assess freeze-drying's impact. In conclusion, the properties highlighted in your study are not adequately reflected in your product. To contribute new knowledge to the field, additional tests and characterizations are essential for article publication.
Author Response
Reviewer 1
Dear reviewer, first, all authors thanks you for your reviewing and comments on our manuscript.
Please find following our answers to your comments:
- The methodology for leachate preparation appears to be lacking. It does not clearly outline the conditions for digestion. Is the closed vessel the only method used? What was the humidity level during fermentation? Does the leachate contain solids?
As explained in the introduction, Musa paradisiaca rachis leachate is a traditional and well-known agricultural treatment in South America and Cuba. Thus, the leachate was prepared according to the traditional method, with small pieces of Musa paradisiaca rachis (French plantain cultivar) left to decompose in a closed vessel at laboratory temperature, pressure and humidity. The leachate was collected without any solid material after 2 months of decomposition. All these elements are added in the Materials and Methods section.
In my opinion, the use of freeze-drying is not recommended as it increases production costs and complicates scaling. Your results indicate that when you lyophilize the sample, the concentrations of nitrogen, potassium, phosphorus, and calcium decrease. If freeze-drying is necessary, a more thorough explanation of the rationale behind this choice is required.
Freeze-drying was used to try to standardize the process for replicates and to find the biologically active concentration. This explanation has been added to the Materials and Methods section. The lyophilized leachate was also used to characterize by solid-state NMR, which has the advantage of performing the experiment directly on the powder. We showed that there was no significant loss of information compared to the results obtained by liquid state NMR. However, should the extract be used on a large scale, it would be in its liquid form, thus minimizing production costs.
- It is crucial to standardize the leachate preparation process. Consider conducting a statistical design to determine the variables that have the most significant impact on leachate quality. Since your article is focused on leachate, meticulous attention to the conditions is essential, as these conditions can significantly affect your results.
We couldn't agree more, which is why we decided to use a freeze-drying process after traditional preparation to characterize the leachate and use the same leachate for all chemical, fungicide, biostimulant and elicitor assays.
- There is no reference to the conditions for in vivo experiments in tomato seedlings.
Lines 237 to 240, we indicated that in the phytotron assay, plants were cultivated on Jiffy Growblock at 24 °C (60% relative humidity (RH), 16- to 8-hour day/night cycle), in a Strader® Phytotron for 45 days with irradiance provided by LEDs with growth/flowering spectrum AP67.
- Regarding the gene analysis, did you design the primers, and if so, what criteria guided your primer selection and analysis conditions? Additionally, please provide information about the primer manufacturer.
Primers were designed using (1) a BLAST of the homology transcript sequences of two other plant species (Arabidopsis thaliana and Populus trichocarpa) on the tomato genome; (2) the use of NCBI's Primer-BLAST tool and then (3) produced by Kaneka Eurogentec SA. These precisions were added Materials and Methods section.
- Your leachate does not exhibit fungicidal activity, contrary to your declaration. It is evident from your findings that after 7 days, fungi growth resumes. To substantiate claims of fungicidal activity, additional antagonism tests against different phytopathogenic microorganisms are needed, ensuring that the effect remains consistent over time.
We indicated several times in results section that the leachate has only a fungistatic activity and that after 7 days, the leachate lost its ability to reduce fungal growth.
- A thorough discussion is needed to explain the reasons behind the fluctuations in nitrogen, carbon, potassium, and phosphorus levels.
Freeze-drying is an accepted method for preserving the quality of nutrients in products. However, leachates, which are rich in humic substances with carboxyl groups, are likely to form complexes with metal cations in solution. These complexes may be more or less stable depending on the experimental conditions (pH, ionic strength). It can be assumed that during the freeze-dried process, the organization of the initially free ionic molecules in solution is reorganized. Placing the freeze-dried leachate in aqueous solution for mineral analysis was not sufficient to extract the minerals from the matrix, which can explain the lower results observed compared to the content in liquid leachate (Tipping, 2009. Cations biding by humic substances, https://doi.org/10.1017/CBO9780511535598; Croué et al., 2003. Characterization and Copper Binding of Humic and Nonhumic Organic Matter Isolated from the South Platte River: Evidence for the Presence of Nitrogenous Binding Site, https://doi.org/10.1021/es020676p). Another possible explanation is that after freeze-drying, microbial activity is reduced and C and N levels remain high. In the leachate, however, microorganisms continue to mobilize C and N.
- The article should delve into the types of phenolic compounds that may be present in the leachate. Consult existing literature to identify previously reported compounds and their mechanisms. Confirm these predictions through Mass Spectrometry (MS) analysis.
Untargeted mass spectrometry and NMR were used to investigate the chemical composition of the leachate in a non-exhaustive manner. The benzoic acid detected by NMR and mass spectrometry has already been described in Gmelina arborea leaf leachates (Madhan Shankar et al., 2014. https://doi.org/10.1155/2014/108682), and Northern and Central European trees (beech, birch, oak, ash, common maple, hornbeam, hazelnut, willow, poplar, hawthorn, pine, spruce-fir, douglas-fir and larch) (Kuiters & Sarink 1986. https://doi.org/10.1016/0038-0717(86)90003-9). Among the 1200 compounds predicted by untargeted mass spectrometry, we decided to focus only on the annotated compounds whose activities reported in the literature were matching with those tested in this study and in harmony with the NMR results.
- Please provide an explanation as to why leachate loses its fungicidal activity when sterilized using a 0.22 µm filter. Additionally, discuss the stability of liquid leachate compared to freeze-dried leachate.
The fungistatic activity loss of the lixiviate after sterilization by a 0.22 µm filtration could be explained by the retention of microorganisms but also by the adsorption losses of chemical compounds on membrane filters both with fungistatic activity. Some compounds like salicylic and benzoic acids were reported to show significant adsorption losses on membrane filters (Carlson & Thompson, 2000. https://doi.org/10.1093/chromsci/38.2.77).
The microbiota of the leachate is unknown, and it can be assumed that some of them with fungistatic activity might be sensitive to freeze-drying and be inactivated.
- It is worth noting that your leachate negatively affects the root length of tomato seedlings, which may impact the mobilization of nutrients and minerals. Consider conducting viability experiments in a greenhouse setting to further assess freeze-drying's impact.
The experiments were performed in drought stress. It is well known that in drought stress, the root length was increased (Farooq et al. 2009. https://doi.org/10.1051/agro:2008021). Thus, the reduction of root length is a positive effect of application of the leachate as a biostimulant indicating a reduction of the stress perceived by the plant. This element was added in Results and discussion section.
Reviewer 2 Report
In this paper, the authors focus on Musa paradisiaca leachate and evaluate the chemical composition and mineral contents. The biological potential roles as fungicides, plant defense elicitors, and plant biostimulants were also tested. Overall, the manuscript is well-organized and written. It should be published after minor revisions.
- For the chemical composition of leachate, I’m wondering if the authors run a fingerprint and try to purify some of the main peaks and use it for the latter activity testing.
- Table 1, why the content of Mg is zero in freeze-drying leachate? Is it detection sensitivity? How many times did you repeat the experiments?
- In Figure 1, please clarify the meaning of the alphabet above each column.
Author Response
Reviewer 2
Dear reviewer, first, all authors thanks you for your reviewing and comments on our manuscript.
Please find following our answers to your comments:
In this paper, the authors focus on Musa paradisiaca leachate and evaluate the chemical composition and mineral contents. The biological potential roles as fungicides, plant defense elicitors, and plant biostimulants were also tested. Overall, the manuscript is well-organized and written. It should be published after minor revisions.
- For the chemical composition of leachate, I’m wondering if the authors run a fingerprint and try to purify some of the main peaks and use it for the latter activity testing.
Trying to purify and chemically characterize every compound in the leachate is a very exciting project, but also a job in itself that may take several years. Characterizing the biological activity of each purified compound is also a very interesting challenge, but a job in itself that may take several years. This is not the option we have chosen for this publication. More than focusing on the potential activity of one compound in particular, the aim of this study is to use the whole extract/ leachate knowing that synergistic effects can take place. Moreover, the use of the whole extract can help minimizing the production cost due to separation and other treatments.
- Table 1, why the content of Mg is zero in freeze-drying leachate? Is it detection sensitivity? How many times did you repeat the experiments?
There are no Mg in the freeze-drying leachate. Analyses were made in triplicate.
- In Figure 1, please clarify the meaning of the alphabet above each column.
From the output of the Kruskal-Wallis’s test, we know that there is a significant difference between groups, but we do not know which groups are different. The multiple comparison with Dunn’s test identifies the groups with superscript letters. Treatments without common superscript letters (a, b, c, d) differ significantly. We clarified this in the legend.
Round 2
Reviewer 1 Report
1. In accordance with the leachate preparation methodology, I accept your stipulated conditions. However, for forthcoming publications, I strongly recommend employing a statistical design to ascertain the variables that exert the most significant influence on leachate quality and to standardize the process. Given the focus of your article on leachate, meticulous attention to these conditions is imperative, as they can significantly impact the integrity of your results.
2. I would like to reiterate the need for a more comprehensive explanation regarding your choice of freeze-drying.
3. Once again, I must point out that your leachate does not demonstrate fungicidal activity, in contrast to your assertion. It is evident from your findings that after 7 days, fungal growth resumes. To substantiate claims of fungicidal activity, additional antagonism tests against various phytopathogenic microorganisms are required, ensuring consistency over time. Typically, biofungicide activity is evaluated after 5 or 7 days, as a one-day assessment is insufficient, as the observed effects may result from the normal growth of the C. gloesporoides strain.
4. In reference to Figures 2 and 3, it is necessary that you provide an explanation for the insignificance of the variables in question. It is necessary to report the values for both treatments and employ statistical analyses to determine the significance of these treatments.
Author Response
Reviewer 1
Dear reviewer, first, all authors thanks you for your reviewing and comments on our manuscript.
Please find following our answers to your comments:
- In accordance with the leachate preparation methodology, I accept your stipulated conditions. However, for forthcoming publications, I strongly recommend employing a statistical design to ascertain the variables that exert the most significant influence on leachate quality and to standardize the process. Given the focus of your article on leachate, meticulous attention to these conditions is imperative, as they can significantly impact the integrity of your results.
We note with interest the advice.
- I would like to reiterate the need for a more comprehensive explanation regarding your choice of freeze-drying.
As indicated in the manuscript, the freeze-drying was used to find the biological active concentration and standardize the procedure for the assays. Moreover, freeze dried leachate was used to be characterized by solid state NMR which presents the advantage of performing the experience directly on the powder. We showed that there was no significant loss of information in comparison to the results obtained by liquid state NMR. However, should the extract being used at a large scale, it would be in its liquid form, minimizing indeed production costs.
- Once again, I must point out that your leachate does not demonstrate fungicidal activity, in contrast to your assertion. It is evident from your findings that after 7 days, fungal growth resumes. To substantiate claims of fungicidal activity, additional antagonism tests against various phytopathogenic microorganisms are required, ensuring consistency over time. Typically, biofungicide activity is evaluated after 5 or 7 days, as a one-day assessment is insufficient, as the observed effects may result from the normal growth of the C. gloesporoides strain.
As you can see on table 3 and Figure 1, the effect of the leachate cannot result of the normal fungal growth because the control does not induce any inhibition. Fungistatic activity is a common concept well known in medical microbiology (Lewis, J. S., & Graybill, J. R. (2008). Fungicidal versus fungistatic: what's in a word? Expert opinion on pharmacotherapy, 9(6), 927-935). Fungistatics are anti-fungal agents that inhibit the growth of fungus (without killing the fungus) and/or showed very rapid regrowth of the pathogen. However, to avoid any confusion, the word 'fungicide' in the text have been replaced by 'fungistatic' in all the manuscript.
- In reference to Figures 2 and 3, it is necessary that you provide an explanation for the insignificance of the variables in question. It is necessary to report the values for both treatments and employ statistical analyses to determine the significance of these treatments.
Values and p-value were added in Figure 2 and supplemental Figure S4.